# A Comprehensive Overview of Neurophysiological Correlates of Cognitive Impairment in Amyotrophic Lateral Sclerosis

**DOI:** 10.3390/cells15010037

**Published:** 2025-12-24

**Authors:** Seyyed Bahram Borgheai, Brie E. Achorn, Alyssa H. Zisk, Sarah M. Hosni, Karl E. G. Richter, Frank S. Menniti, Yalda Shahriari

**Affiliations:** 1Department of Electrical, Computer & Biomedical Engineering, University of Rhode Island, Kingston, RI 02881, USA; borgheai@emory.edu (S.B.B.); shosni@odu.edu (S.M.H.); 2Neurology Department, Emory University, Atlanta, GA 30322, USA; 3Interdisciplinary Neuroscience Program, University of Rhode Island, Kingston, RI 02881, USA; briennaachorn@uri.edu (B.E.A.); alyssazisk@uri.edu (A.H.Z.); 4AssistiveWare B.V., 1016 PL Amsterdam, The Netherlands; 5Computer Science Department, Old Dominion University, Norfolk, VA 23529, USA; 6IMS Scientific Consulting, Malden, MA 02148, USA; krichter7.mail@gmail.com; 7MindImmune Therapeutics, Inc., Kingston, RI 02881, USA

**Keywords:** amyotrophic lateral sclerosis, cognition, neurophysiology, biomarkers

## Abstract

Amyotrophic lateral sclerosis (ALS) is a progressive neurodegenerative disease that leads to the gradual loss of motor control, typically resulting in paralysis and death within 3 to 5 years of diagnosis. ALS shares neuropathological and genetic associations with fronto-temporal dementia (FTD), a neurodegenerative condition primarily impacting cognitive functions. These two conditions are increasingly viewed as manifestations of a single molecular disease process that affects distinct brain systems, impacting motor neuronal pathways in ALS and fronto-cortical functions in FTD. However, this simple dichotomy belies the complexity of these conditions. In particular, patients with primary motor ALS can also experience significant cognitive deficits. Investigating the pathobiological and neurophysiological underpinnings of these impairments is essential for a comprehensive understanding of ALS and may open avenues for targeted therapies to alleviate these debilitating symptoms. Moreover, the biophysical correlates of cognitive deficits in ALS may serve as sensitive biomarkers for evaluating potential therapeutics. In this narrative review, we begin with an overview of the clinical features and genetics of ALS, followed by a review of the associated cognitive deficits that are adjunctive to motor decline. We then highlight neuroimaging studies from our laboratory and the broader literature, using EEG and other modalities that are beginning to uncover systems-level brain disruptions potentially underlying cognitive impairment in motor-dominant ALS.

There is emerging evidence that amyotrophic lateral sclerosis (ALS) extends beyond a purely motor disorder to involve significant cognitive and systems-level brain dysfunction. This narrative review situates ALS within the broader disease spectrum, highlighting shared neuropathological, neurophysiological, and genetic mechanisms while emphasizing the substantial heterogeneity present even in motor-dominant ALS phenotypes. We synthesize clinical and genetic insights alongside electrophysiological and neuroimaging evidence, drawing on complementary modalities from our laboratory and the broader literature, to elucidate neural network disruptions that may underlie cognitive impairments in ALS. By integrating these findings, the review underscores the potential of cognitive biomarkers for disease (sub)characterization and therapeutic applications.

## 1. Search Method

This is a narrative review that draws on literature identified through targeted searches in PubMed, IEEE Xplore, and Google Scholar, as well as relevant book chapters. The search focused on correlates or biomarkers of cognitive dysfunction in ALS with topics related to clinical, genetic, cognitive screens, structural and functional neuroimaging. We also targeted articles adopting multimodal neuroimaging, e.g., EEG-fNIRS integration, particularly with cognitive tasks performed by ALS patients. Given the narrative nature of this review, the search was not exhaustive; instead, studies were selected based on conceptual relevance and contribution to abovementioned major contexts.

## 2. Clinical Features of Amyotrophic Lateral Sclerosis (ALS)

In 1869, French neurologist Jean-Martin Charcot devised the term “amyotrophic lateral sclerosis” to represent a clinical syndrome of muscle atrophy and weakness (amyotrophic), along with tissue scarring in the lateral spinal cord (lateral sclerosis), resulting in paralysis and death [1]. ALS is now recognized as a progressive neurodegenerative disorder that primarily affects adults between 40 and 70 years of age, with peak onset in the mid- to late-50s [2]. The defining motor symptoms of ALS are caused by the degeneration of upper motor neurons in the motor cortex and corticospinal tracts, along with lower motor neurons in the brainstem and spinal cord, leading to progressive weakness, loss of voluntary motor control, and eventual paralysis [2,3,4]. The disease typically begins with focal weakness that then spreads to include most muscles, with the failure of eye, sphincter, and diaphragm muscles occurring in the late stages of the disease [2]. A diagnosis requires a comprehensive evaluation of clinical symptoms, neurological examinations, electromyography, and lab tests, such as blood and genetic testing [4]. Classic ALS constitutes 70% of all ALS cases and can be categorized into limb-onset, bulbar-onset, or respiratory-onset, depending on the site of initial symptom manifestation [5]. These categorizations notwithstanding, ALS is a heterogeneous disease, with significant individual variability in initial presentation, disease progression, spreading patterns, and survival [2]. This heterogeneity relates to the diverse pattern of upper and lower motor neuron involvement among patients [1]. Like other complex diseases, there is a diagnosis continuum with ALS; this means that patients may be given a diagnosis of suspected, possible, probable, or definite ALS [4]. Ultimately, a diagnosis of definite ALS is given when the clinical examination and electromyography indicate both upper and lower motor neuron signs in three body regions, and laboratory tests rule out all other diseases that may be mistaken for ALS. Today, it is estimated that approximately 3 to 5 per 100,000 individuals will develop ALS [4]. Patients can be classified based on whether there is a family history of the disease, which accounts for ~10% of cases and is known as familial ALS (fALS). However, for most patients, the cause of ALS is unknown, and no family history is present; these cases are referred to as sporadic ALS (sALS) [2].

## 3. Genetics

Insight into possible molecular mechanisms of pathogenesis in ALS has come from the discovery of genetic variants that lead to ALS in familial cases. Furthermore, genome-wide, population-based studies suggest that as much as 50% of the risk for “sporadic” ALS may have a genetic basis [6]. To date, variations in over 30 genes have been linked to ALS [3]. Of these, Superoxide dismutase [Cu-Zn] (SOD1), the guanine nucleotide exchange factor C9ORF72, Transactive response DNA-binding protein 43 (TDP-43), and Fused in sarcoma (FUS) are the most well-documented, accounting for two-thirds of fALS cases and ~10–15% of sALS cases (Table 1) [7,8]. It is now established that a patient’s genetic background can greatly influence clinical phenotypes. For instance, individuals with SOD1 mutations are more likely to exhibit limb-onset ALS, while those with C9ORF72 mutations are more likely to present with bulbar-onset ALS and to develop psychiatric disorders [9,10]. What is known about mechanisms relating genetic variations to disease phenotypes is outlined below.

SOD1 was the first ALS gene to be identified in 1993 and remains one of the most studied genetic contributors [11]. Individuals with SOD1 mutations typically experience symptom onset in the lower limbs and lower motor neurons, but the clinical phenotype varies across different SOD1 variants [11]. For example, the Gly73Ser and Ser106Leu mutations are associated with a more severe phenotype, while patients homozygous for the Asp91Ala allele have a slowly progressing disease course with urinary incontinence and sensory abnormalities appearing in the late stages [12]. SOD1 is ubiquitously expressed and acts as a potent antioxidant that protects cells from superoxide radicals [13]. SOD1 mutations are thought to result in a gain-of-toxic-function due to protein misfolding and aggregation. Mutations may also alter protein function, leading to aberrant protein–protein interactions. The accumulation of toxic hydroxyl radicals can damage both nuclear and mitochondrial DNA [13,14].

The most common genetic cause of ALS is a G_4_C_2_ hexanucleotide repeat expansion in the C9ORF72 gene, accounting for ~34% of fALS cases [15]. In the CNS, neurons and microglia express the highest levels of C9ORF72, while myeloid cells, particularly innate immune dendritic cells, show the highest expression in the periphery [16]. The C9ORF72 gene encodes a protein that is believed to function as a Differentially Expressed in Normal and Neoplasia (DENN) domain protein that serves as a GDP/GTP exchange factor (GEF) to activate Rab-GTPases involved in membrane trafficking [17]. C9ORF72 mutations often lead to adult-onset ALS with deficits in RNA metabolism, autophagy, immune modulation, nucleocytoplasmic transport, and protein homeostasis [11].

TDP-43 is a DNA/RNA-binding protein that regulates the expression of thousands of genes and seems to be particularly important for RNA metabolism [15,18]. TDP-43 is ubiquitously expressed and tends to localize to discrete subnuclear structures [19]. In 2006, it was observed that TDP-43, encoded by the TARDBP gene, exhibited a pattern of nuclear depletion and subsequent cytoplasmic accumulation in the motor neurons of ALS patients [20]. Individuals with mutations in TARDBP typically present with adult-onset ALS and demonstrate slow disease progression; however, those with certain variants, such as G376D, can exhibit rapid progression [11]. These mutations tend to lead to defects in RNA metabolism, stress granule formation, and axon pathology.

FUS is a DNA/RNA-binding protein that plays an important role in DNA repair and RNA metabolism [21]. Mutations in FUS have been linked to ALS, with over 50 different variants observed in ALS patients [22]. Corroborating this, FUS mutations lead to impairments in RNA metabolism, stress granule formation, DNA repair deficits, and axon pathology [11]. Individuals with ALS-linked FUS mutations exhibit juvenile-to-adult onset, with an aggressive disease course and cognitive impairment and postural tremors in some patients [11].

## 4. Cognitive Deficits in ALS

Although ALS is defined as a disease of motor pathways in the central nervous system (CNS), other brain regions can be affected and other symptoms may be present, notably including deficits in cognitive function [23]. Beyond the motor neurons and muscles, “non-motor” regions of the CNS—including the frontal, parietal, and somatosensory areas—can be affected, contributing to the neuropsychiatric and cognitive symptoms commonly observed in patients with the disease [24]. Determining the nature and frequency of non-motor, specifically cognitive, symptoms, though, can be technically challenging. Modified evaluation tasks have been proposed to make evaluations accessible to patients with motor and speech impairments [25], but this may affect interpretation, as norms for cognitive evaluations typically do not include people with significant motor or speech impairments. Rapid declines make longitudinal studies difficult due to the increasing inaccessibility of evaluations and lack of applicable norms as disability severity increases. Mood disorders—whether stemming from underlying disease biology, coexisting conditions, or psychological reactions to a fatal and currently incurable illness—are known to impact the assessment of cognitive functions. However, studies have varied in their use of cognitive evaluation methods and diagnostic tools. Cognitive function in ALS patients is commonly assessed using general tools such as the Montreal Cognitive Assessment (MoCA) and the Mini Mental State Examination (MMSE). ALS-specific instruments have also been developed. The Edinburgh Cognitive and Behavioural ALS Screen (ECAS) or ALS Cognitive Behavioral Screen (CBS), designed specifically for people with ALS [26], has been validated in several populations, including in the UK, where it was first developed, as well as in Italy [25,27] and China [28]. Additionally, many studies apply the revised Strong criteria to determine the presence or absence of ALS-cognitive impairment (ALSci) [28].

The prevalence of cognitive deficits in ALS has been investigated in a number of studies. A population-based study [29] in an Irish cohort of ALS patients (*n* = 160 patients, 110 matched controls, excluding those patients with frank dementia) found 34% had cognitive impairment (as defined by ECAS) while 47% had no detectable cognitive deficits. In another comprehensive study [30], in non–speed-dependent cognitive tasks, cognitive impairment was observed in 51% of patients with ALS (*n* = 279), compared with only 5% of controls (*n* = 129); notably, 15% of patients met diagnostic criteria for FTD. A similar study [31] in a geographically distinct cohort in Italy found very similar results: of the 188 ALS patients that were evaluated, approximately 30% had at least one category of cognitive impairment, 50% were cognitively normal, and 13% had ALS-FTD. Results were very similar in a study using the ECAS in China: about 35% of ALS patients had cognitive impairment [32]. In a recent, longitudinal study of ALS patients (*n* = 189 at start of study), Costello et al. [33] found that 23% of all patients met criteria for cognitive impairment at the first assessment (mean time from symptom onset to evaluation was 19 months). On average, overall cognition was relatively spared over the course of three follow-up sessions over one year. However, by eight months, only 42% of starting patients remained in the study, dropping to 26% by one year. Costello et al. [33] attempted to control for non-random dropout using Cox survival models accounting for age, diagnostic delay, site of onset, and C9orf72 genotype, but with only 49 of 189 patients assessed at the last follow-up, it is unclear whether the statistical analysis was sufficient to correct for the attrition rate. Meta-analysis has also been used to evaluate the presence of cognitive deficits over a greater number of patients and to pool results across geographic regions. A recent meta-analysis [34] of 45 cross-sectional studies and 13 longitudinal studies found results similar to those of Costello with respect to the percentage of patients with cognitive impairment at diagnosis. The number of patients with cognitive impairment at the start of the study ranged from 14% to 54%, with a weighted average of 37%. Thus, despite the different populations and methods, approximately one-third of ALS patients already suffer from some form of cognitive impairment by the time of diagnosis.

While the presence of cognitive deficits in ALS patients is reasonably well established, the natural history of cognitive decline is less clear. An influential study by Elamin and colleagues indicated a decline in cognition over time [35]. This study further noted that the population with cognitive impairment at diagnosis also suffered from a more rapid progression of other symptoms of ALS, including motor deficits. These investigators developed an algorithm using a simple score that included (1) site of onset (bulbar vs. spinal), (2) rate of decline measured using the ALS Functional Rating Scale (Revised), and (3) impairment of executive function (but not ECAS) that had predictive value for the prognosis of disease in two test cohorts [23]. Information on patient genotype for C9orf72, FUS, TARDBP (TDP-43) and optineurin, available for nearly all cases, was incorporated in a subsequent analysis and indicated a higher percentage of C9orf72 expansion carriers in the group of patients suffering the most rapid declines. However, in contrast to Elamin and colleagues’ results, two other groups using a similar or greater number of patients did not see a progressive loss of cognitive abilities over time [25,36]. Meta-analyses have also been performed to help clarify progression of cognitive decline in ALS. Benbrika et al. [37] reviewed 190 articles from the English and French language literature and found ample evidence for cognitive deficits at diagnosis, and while there was some evidence of worsening toward the end of the disease, there was not consistent evidence of decline throughout the disease. More recently, a somewhat overlapping meta-analysis by Finsel et al. [34] found similar results. This meta-analysis found some studies in which a progressive decline was present but also many where a decline was not detected despite sufficient numbers of patients to observe a decline had it been present. Both the review and the meta-analysis pointed to several factors that could confound either longitudinal studies or cross-sectional studies with age-matched controls. Depression is worse shortly after diagnosis, and that depression typically reduces measured cognitive performance, thus possibly lowering apparent cognitive power early in the disease, masking any later true decline in cognitive ability. There are also practice effects when using the ECAS repeatedly, possibly artifactually increasing cognition scores in retests [25].

Taking into consideration the limitations of prior work, a recent, large study to assess cognitive function by McHutchison et al. included whole-genome sequencing for each patient and repeated cognitive and behavioral evaluation every 3–6 months for up to six total assessments using ECAS and structured interviews [38]. Three different versions of ECAS were used to reduce practice effects. Latent class growth analysis, a statistical procedure similar to principal component analysis, was used to identify subgroups of cognitive phenotypes within the study population over time [39]. Their outcomes showed that lower ECAS scores at baseline were correlated with a decrease in cognition over time. Furthermore, they found that C9orf72 expansion carriers were over-represented in this group, but C9orf72 status alone did not fully explain progressive deficits in cognition.

In summary, and as concluded by Finsel et al. [34], there is strong evidence for cognitive dysfunction early in the course of ALS, possibly even before diagnosis, and this could be considered a characteristic of ALS or of subtypes of ALS. However, the evidence for a gradual, progressive decline in cognition is weaker [34]. While it does occur in some cases, there is insufficient evidence for a uniform deterioration of cognition in all ALS patients as the disease progresses. Several authors emphasized that testing and following presymptomatic carriers of ALS genes may be a way to determine whether early decreases in cognition may be part of the disease process in those groups [34,40].

## 5. Brain Structural Correlates of Cognitive Decline in ALS

Postmortem studies of ALS neuropathology have shown that even when only lower motor neuron symptoms were present, mis-localized, aggregated TDP-43 can be found in many parts of the CNS, not only motor pathways [41]. Post-mortem evidence of synapse loss in the prefrontal cortex has been correlated with cognitive decline during life in ALS patients [42]. Thus, both positive (TDP-43 mis-localization and/or accumulation) and negative (synapse loss) elements of neuropathology have been demonstrated in CNS regions beyond motor pathways in ALS.

There is also evidence of degeneration beyond the motor cortex during life, although correlating ALS symptom-defined subtypes with neuroimaging markers remains challenging [43]. Cortical thinning has been detected by structural magnetic resonance imaging (MRI) in ALS patients with or without cognitive impairment or FTD-like symptoms [44]. There is some evidence that thinning of the right fronto-temporal insular cortex may be specifically associated with ALS with cognitive impairment as compared with ALS without cognitive deficits [45]. The Canadian ALS Neuroimaging Consortium (CALSNIC) studied the relationship between ECAS scores and MRI-based brain morphometry [46]. In addition to the expected white matter degeneration in the corticospinal tracts, they found degeneration in the corpus callosum, cingulum and superior longitudinal fasciculus of ALS patients with impaired verbal fluency or executive dysfunction. In one of the largest and most comprehensive studies to date of neuroimaging correlates of cognition in ALS patients (293 patients, 237 controls), Tan and colleagues found significant changes in both white and gray matter that correlated with impaired function in specific cognitive domains [47]. Patients were evaluated a maximum of five times at three-to-six-month intervals. The changes were measured by MRI, using cortical thickness and subcortical structural volume of gray matter and connectome analysis of diffusion tensor imaging of white matter. Language deficits were significantly associated with frontal, temporal, parietal and subcortical gray matter neurodegeneration; memory dysfunction with hippocampal and medial-temporal atrophy; and executive impairment with widespread white matter changes. These results establish that structural changes outside the corticospinal tracts can be analyzed by MRI and are correlated, on a population basis, with the types of cognitive losses that can occur in ALS.

Besides MRI, other techniques have been used to investigate possible correlates of extra-motor degeneration in ALS. Regional cerebral blood flow measurements by positron emission tomography (PET) have been used as a proxy for neural activity. In a small study, Abrahams and colleagues compared cognitively impaired ALS patients with unimpaired patients and controls [48]. They found cognitively affected patients displayed significantly impaired activation in cortical and subcortical regions, including the dorsolateral prefrontal cortex, lateral premotor cortex, medial prefrontal and premotor cortices, insular cortex, and the anterior thalamic nuclei. In a much larger study [49], fluorodeoxyglucose positron emission tomography (FDG-PET) identified changes in the brain associated with cognitive decline in ALS patients [50]. They found hypometabolism in the frontal lobe present in ALS patients with cognitive impairment and even greater hypometabolism in ALS-FTD patients. ALS patients without cognitive impairment were not similarly affected. They also found hypermetabolism in the cerebellum of patients with ALS-FTD. Foucher et al. [51] also used FDG-PET and found a correlation between ECAS score and metabolic activity. They identified the right orbitofrontal gyrus as a region of reduced metabolic activity that correlated with poorer ECAS scores. They also found regions of hypermetabolism in the occipital cortex and cerebellum, similar to Canosa and colleagues’ findings.

Although the FDG-PET method provides less spatial resolution than MRI, both methods have detected differences between ALS patients with cognitive dysfunction and both cognitively healthy ALS patients and similarly aged, cognitively healthy controls. Both techniques have identified changes specifically in white and gray matter in selected regions in the frontal lobes, providing converging evidence that cognitive decline in ALS may be correlated with imaging changes in regions generally associated with language and cognition that lie outside the canonical motor pathways associated with ALS neurodegeneration.

## 6. Functional Neuroimaging Correlates of Cognitive Impairment in ALS

As ALS clinical symptoms evolve, the corresponding neural dysfunction may be probed through various electrophysiological and neuroimaging techniques. These techniques include electroencephalography (EEG), which measures the electrical signals at the scalp generated by synaptic activity in the underlying cortex [52]; functional near-infrared spectroscopy (fNIRS), which measures hemodynamic responses associated with cortical activity by detecting changes in oxygenated and deoxygenated hemoglobin^5^; and functional magnetic resonance imaging (fMRI), which measures changes in blood oxygen utilization also resulting from cortical synaptic activity [53,54,55]. While single-modality studies continue to produce significant insights, combining neuroimaging techniques from different modalities is being increasingly employed to provide a more comprehensive view of ALS-related neural alterations [56,57,58,59,60]. Together, such techniques offer promising windows into both the motor and non-motor manifestations of the disease. Here, our focus is on the cognitive neural signatures of ALS as revealed through studies that either used resting-state or task-based paradigms (Table 2) using single or multimodal neuroimaging techniques.

Resting-state functional connectivity (RSFC) has been widely explored in ALS. This approach is attractive because it obviates patients’ difficulties with task-based paradigms [61], particularly at advanced stages of the disease. Increased RSFC has been reported in ALS patients during eyes-closed fMRI, particularly in the default mode (DMN) and frontoparietal (FPN) networks involving prefrontal, frontal, and parietal regions [49,62,63,64]. This hyperconnectivity has been linked to clinical and cognitive impairments [65], disease progression, and areas of structural disconnection [62,63]. Conversely, other eyes-closed fMRI studies have found reduced RSFC in networks associated with cognitive and behavioral functions [66,67,68]. Using EEG, Fraschini and colleagues [69] reported reduced electrical RSFC, while Kopitzki and colleagues [70] found preserved hemodynamic RSFC using fNIRS. A high-density longitudinal EEG study during eyes-open rest identified elevated gamma coherence between fronto-parietal regions and theta coherence across bilateral motor areas [71]. Another study found reduced EEG power in ALS patients across multiple regions, including delta/theta bands in prefrontal areas, beta in sensorimotor, and delta/alpha/beta in occipital and temporal lobes [72]. In eyes-closed studies [59,73,74], alpha power reduction was associated with disease-related structural degeneration, particularly loss of pyramidal neurons. Thus, there is little consensus on the biomarkers identified using single-mode neuroimaging methods. The apparent discrepancies among these various studies may stem from methodological or imaging technique differences that affect RSFC estimates [69,75]. This emphasizes the importance of multimodal approaches to fully characterize ALS-related neural alterations.

Connectivity-focused multimodal resting-state paradigms may help to clarify the apparent conflicts among the single-modality studies. An fNIRS-EEG study by Deligani et al. [59] reported significantly increased fronto-parietal EEG connectivity in the alpha and beta bands, along with enhanced interhemispheric and intra-right hemisphere fNIRS connectivity in frontal regions. Their study also observed reduced EEG spectral power in the alpha and theta bands across frontal, central, temporal, parietal, and occipital regions. Moreover, increased hemodynamic spectral power was observed in the frontal (very low frequency oscillations (VLFO)~0.01–0.05 Hz) and parietal regions (VLFO and low frequency oscillations (LFO) ~0.05–0.15 Hz) in ALS patients. Similar findings were reported by Kopitzki et al. [70], who demonstrated altered functional connectivity using DTI and fNIRS during closed-eyes recordings. These changes suggest that, beyond the motor circuits, ALS significantly affects cognitive and extra-motor networks. The default mode network (DMN), which underlies memory, emotional processing, self-referential thought, and aspects of consciousness [76], has shown consistently increased connectivity in ALS, often correlating with greater disability and faster progression [66,68]. Additionally, executive dysfunction in ALS has been linked to disruptions in the frontoparietal (FPN) and dorsal attention networks (DAN), which regulate attention and cognitive control [77,78,79]. Interpretation of the underlying structural changes that result in the changes in connectivity varies, as these have been linked to both impaired inhibition and compensatory mechanisms for structural decline [63]. While there is yet to be a consensus on resting state correlates of cognitive function in ALS, these analyses clearly point to widespread alteration in functional networks associated with the disease, consistent with clinical findings that ALS patients may experience neurocognitive symptoms beyond motor impairment.

Analysis of EEG temporal changes in response to cognitive-sensory events, specifically in the context of event-related potentials (ERPs), has also been widely explored in patients with ALS. In these activation paradigms, EEG signals are monitored under conditions where subjects either passively or actively process sensory inputs such as a repeated tone. The averaged EEG signal has well characterized temporal components that have been mapped to different epochs of early signal processing (see Figure 1). Abnormalities in ERPs, including the P300 component, as well as earlier components such as P100 and N200, elicited during oddball paradigms, are among the most frequently reported temporal features impaired in ALS patients [58,80,81,82]. P300 responses correspond to cognitive substrates related to attentional demand and working memory, collectively referred to as mental workload [83,84,85]. Decreased amplitudes of the P300 response have been linked to markers of ALS progression [80,81,82]. Given that the P300 component is associated with selective attention, visuospatial filtering, and working memory, the reduced P300 amplitude in ALS may reflect deficits in selective attention. Consistent with this interpretation, Shahriari et al. [86] demonstrated that stronger positive deflections around 220 ms at midline frontal electrodes (Fz, Cz) were associated with better task performance in ALS, suggesting that preserved early attentional engagement contributes to more effective cognitive control in this population.

In addition to attenuated P300 amplitudes, ALS patients have also been shown to exhibit significant trial-to-trial variability in the timing of the P300 component relative to stimulus presentation, referred to as latency jitter [87,88,89,90]. Single-trial P300 latencies, which reflect stimulus evaluation time, have long been studied in cognitive neuroscience and are known to correlate with task-related metrics, including reaction times, particularly in tasks emphasizing accuracy over speed [91,92,93]. However, this association between P300 latencies and task-related metrics can break down in neurotypical individuals exhibiting higher latency jitter or when task demands prioritize speed [91,93]. Latency jitter can be considered a form of neural variability, which is an essential feature for adaptive learning but is often maladaptively elevated in a range of neurological conditions [94,95]. Elevated latency jitter has been documented not only in ALS but also in autism [94], attention deficit hyperactivity disorder [92], schizophrenia [96], depression [97], traumatic brain injury [98], disorders of consciousness [99], and dementia [97], showing the potential cognitive deficits ALS patients share with non-motor neurocognitive disorders.

Later ERP components have also been found to be attenuated in ALS patients in a visuospatial mental task paradigm [58]. Notably, the N400 component, a marker commonly associated with semantic memory processing, was less pronounced in subjects with ALS compared to healthy controls, suggesting potential impairments in semantic processing. Additionally, a late positive component (~650 ms post-stimulus), typically elicited during mental arithmetic tasks in healthy individuals, was significantly diminished in ALS patients [100,101]. This reduced activation may reflect deficits in computational or executive processing in the ALS population. These later ERP components, which are often reported as absent or significantly weakened in ALS patients [101], could represent promising new neurophysiological markers of disease-related cognitive impairment.

While ERP analyses shed light on both early sensory processing and some higher cognitive deficits in patients with ALS, spectral measures have shown potential to reflect a broader spectrum of cognitive and extra-motor deficits in higher-order cognition using more complex tasks, including motor imagery, the mental execution of a motor action without actual physical execution. Brain systems activated during motor imagery include those associated with physical execution through integration in a system referred to as the mirror neuron system. Some studies [102,103] suggest that brain regions classically associated with physical speech production and language use, namely Broca’s and Wernicke’s areas, are integrated into the human mirror neuron system [104] and are activated during motor imagery tasks [105,106]. Broca’s area has also been found to contain a motor representation of hand actions [107]. Motor imagery is associated with changes in the synchrony of neural activity across brain regions, particularly in the mu (8–12 Hz) and beta (13–25 Hz) frequency bands. Both amplitude suppression (event-related desynchronization, ERD) and enhancement (event-related synchronization, ERS) in these bands have been observed at the onset of a motor imagery task [108,109,110,111]. Studies have found correlations between ERD/ERS amplitudes and clinical measures of ALS progression, such as upper limb and respiratory function [112,113]. Dysfunctions in imagery-specific cortical synchronization have been linked to impaired sensory-motor coupling in the somatosensory cortex [114]. This interpretation aligns with earlier fMRI findings [115], which demonstrated reduced cortical activation in ALS patients during motor imagery tasks. One proposed explanation for reduced ERD during motor imagery is the disruption of the prefrontal-parietal network involved in imagery tasks, an effect possibly stemming from ALS-related extra-motor pathology involving the prefrontal cortex, as suggested by previous neuroimaging and neuropathological studies [116,117,118]. In a separate study, Kasahara and colleagues suggested that reduced ERD in ALS may be a consequence of progression-related neuronal loss based on a quantitative EEG (QEEG) resting-state analysis [112]. The authors speculated that this might reflect an association between bulbar deficits and cognitive dysfunction, potentially due to fatigue or attentional deficits. Further, the reduced ERD patterns observed in ALS patients may reflect mirror neuron system dysfunction, as it has been proposed that suppression in both the mu and beta frequency bands may serve as an index of mirror neuron system activity [119,120].

Other spectral changes have been observed in subjects with ALS in non-motor tasks. In a brain–computer interface (BCI) study employing a visuo-mental paradigm that integrated a P3Speller with mental arithmetic, Borgheai et al. [58] reported a significantly greater increase in frontal delta power from baseline in healthy controls than in ALS patients. Delta power augmentation during mental calculation tasks is generally interpreted as reflecting heightened internal concentration, facilitating the suppression of external interference during cognitive performance. The attenuated delta power increase observed in ALS patients is therefore speculated to reflect diminished internal concentration during task execution. Notably, this electrophysiological signature was also correlated with lower cognitive scores in the ALS group, further supporting its potential as a marker of cognitive dysfunction in this population. A similar pattern was reported in a longitudinal BCI study of individuals with ALS conducted by Shahriari and colleagues [86], where higher delta activity at occipital locations was negatively correlated with daily BCI performance, reinforcing the link between excessive delta activity and impaired attentional or cognitive processing. In the theta frequency band, healthy participants showed a profound power increase, while this pattern was attenuated significantly in the ALS patient group [58]. A smaller increase in theta power was also associated with lower cognitive scores in ALS patients. Theta oscillations are generally linked to multiple cognitive domains, including attentional engagement and working memory processes [58,121]. Thus, elevated theta activity during the mental task may reflect task-imposed increased cognitive workload and attention. Additionally, a significant reduction in beta power was observed in the healthy control group compared to ALS patients. Frontal beta activity is typically associated with top-down control mechanisms [122,123], particularly related to general task-related processing. Beta power is known to increase during the intended maintenance of a “status quo” mental state [124], while its later suppression is thought to reflect content-specific frontal modulation related to working memory and decision-making, as supported by recent studies [125,126]. In agreement, Shahriari et al. [86] showed that higher beta power at occipital electrodes was positively correlated with successful BCI use in people with ALS, suggesting that preserved beta modulation may contribute to sustaining attentional control and visual processing in demanding tasks. Importantly, significant correlations were found between scores on the CBS- and EEG-derived delta and theta power [118], which show the clinical relevance of these biomarkers, though their causal mechanisms are yet to be known.

Beyond basic spectral-temporal features, neuronal network dysfunction in ALS also appears to be accompanied by dysregulation in neurovascular coupling [57,127,128]. Using a multimodal EEG-fNIRS recording with a visuo-mental protocol, Borgheai et al. [58] showed predominantly positive associations in EEG–fNIRS correlation maps in healthy controls, while participants with ALS exhibited largely negative correlations, indicating a desynchronization between electrical (EEG) and hemodynamic (fNIRS) responses during mental tasks. Interestingly, significant hemodynamic differences were localized to fNIRS locations corresponding to the dorsolateral prefrontal cortex (DLPFC), a region known to be critically involved in working memory processes [129,130,131]. This finding further supports the hypothesis that individuals with ALS may exhibit executive dysfunctions affecting cognitive workload processing, particularly within the DLPFC. Moreover, a recent multimodal BCI study [57] demonstrated that integrated EEG–fNIRS features recorded during a brief pre-screening session could predict which workload levels in a subsequent visuo-mental BCI task would yield higher performance. Using multivariate statistical models, the authors showed that combined CBS battery measures (e.g., attention and concentration), hemodynamic features (such as the range of HbO_2_ responses across frontal fNIRS channels and fronto-temporal RSFC), and resting-state EEG spectral markers (including temporal–parietal delta changes) collectively predicted variability in BCI performance.

**Figure 1 cells-15-00037-f001:**
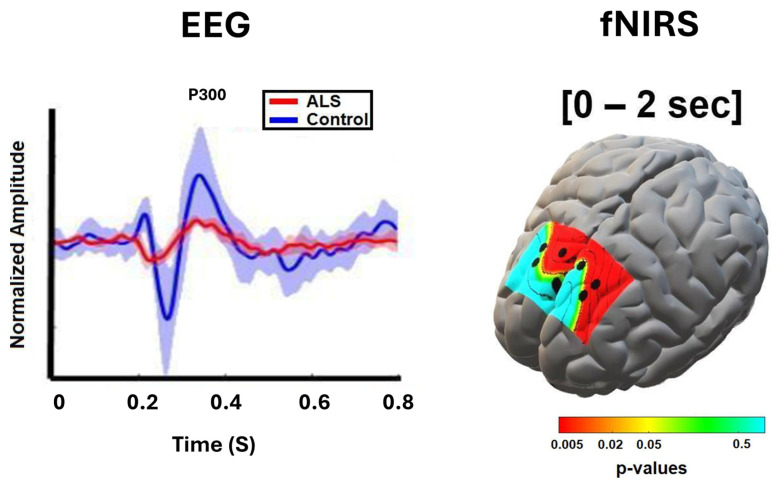
(**Left**): Normalized target ERPs averaged across all ALS participants (red) and all healthy controls (blue) in the 800 ms following stimulus onset in Parietal-occipital region. The shaded bands (light blue and red) indicate the standard deviation within the control and patient groups. (**Right**): *p*-value maps derived from statistical comparison of peak values of hemodynamic concentrations in 0–2 s post-stimulus (adapted from Borgheai et al. [127]).

The data reviewed above indicate that neural signaling abnormalities are observed at multiple levels in ALS. Recent studies have used graph theory to analyze frontal network dysfunctions in ALS patients compared to healthy controls. Graph theory is emerging as a promising framework to analyze the connections between measured network disruptions and observed functional neurocognitive deficits. Graph theoretical analyses model the brain as a network where recording channels or regions of interest (ROIs) act as nodes and their functional or structural connections serve as links [132,133]. These models are then mapped back to the roles of the nodes and their connections to cognitive processes. Sorrentino et al. [133] used magnetoencephalography (MEG)-based minimum spanning tree (MST) analysis to report a shift toward a more centralized and randomized network configuration across all frequency bands as the disease progressed. This pattern has similarly been observed in MST-based analyses of other neuropsychiatric conditions, such as schizophrenia [133] and major depressive disorder [134]. Building on these applications, Hosni and colleagues [60] introduced a nonlinear dynamic graph-based framework using recurrence quantification analysis (RQA) features extracted from EEG and fNIRS. By applying nonlinear graph-derived measures, they demonstrated that these approaches could capture subtle and complex patterns of brain dynamics that are often missed by conventional spectral analyses. This work highlights the potential of integrating nonlinear graph metrics with traditional graph-theoretical models to provide a more comprehensive characterization of network alterations in ALS, particularly in relation to disrupted connectivity and abnormal information flow.

Complementing these findings, an activity-based fNIRS study using a visuo-mental protocol also revealed a more centralized frontal network in ALS patients compared to controls [127]. This star-like organization—marked by increased maximum degree and betweenness centrality—suggests a shift toward a more random, dysregulated structure [134,135], where information flow becomes overly reliant on a few central hubs. Furthermore, Borgheai et al. [136] performed nodal analyses to characterize key network hubs, identifying the right prefrontal cortex (PFC) as housing the most dominant relay stations in ALS patients—nodes critical for intermodular communication. These hubs are known to be particularly vulnerable to neurodegeneration [137]. Compared to controls, ALS patients exhibited reduced node efficiency and strength in the left PFC, alongside an increase in the right PFC, suggesting a lateralized shift in prefrontal engagement. This rightward asymmetry aligns with earlier neuroimaging findings: PET studies during verbal fluency tasks [48] and ERP studies using dual spatial and working memory tasks [138] also showed altered PFC activity in ALS patients.

These findings converge on the hypothesis that ALS-related executive dysfunction, especially dysfunction involving working memory manipulation, is reflected in altered PFC topology and lateralization. The PFC is a well-established node in executive functioning and working memory networks [139,140], and hemodynamic activity in this region is frequently observed during working memory engagement [58,141,142]. The observed dominance of right PFC hubs in ALS patients may indicate compensatory executive control mechanisms, particularly for functions like inhibitory regulation [143] and memory retrieval monitoring [144]. The right PFC is also critically involved in spatial working memory [145], supporting the notion that hyperconnectivity in this region may reflect difficulty in processing the spatial aspects of dual-task paradigms. Frontal lateralization has been observed in other neuropsychiatric disorders, such as schizophrenia [146] and MDD [134], and the similarity in topological markers may be especially relevant considering the prevalence of depressive symptoms in ALS [147,148,149].

The functional network alterations observed in ALS putatively reflect structural atrophy localized to frontal regions, especially evident in task-negative resting-state fMRI studies [63,150]. Dimond et al. [150], using resting-state MRI-DTI, reported associations between verbal fluency errors and disrupted clustering coefficients in right frontal and temporal lobes, implicating reduced white matter integrity in executive dysfunction. Similarly, Tedeschi et al. [66] documented suppression in the right fronto-parietal network, potentially driven by right-lateralized frontal atrophy. Shahriari et al. [136] reported a significantly broader distribution of high-beta band connectivity and absence of distinct connections in directivity analysis in ALS patients compared to controls during a P300-speller task. Their results possibly reflect the involvement of inhibitory interneuron networks associated with beta-frequency oscillations [63,151,152,153]. One interpretation is that such hyperconnectivity may serve a compensatory role in the context of progressive structural degeneration [65], potentially due to the loss of inhibitory interneurons, a hallmark of ALS pathology [154]. These localized atrophies may undermine several key cognitive executive networks, including the executive control network (ECN), fronto-parietal network (FPN), and dorsal attention network (DAN), all of which encompass frontal regions and could contribute to the broad cognitive impairments observed in ALS [66,72,77,155].

**Table 2 cells-15-00037-t002:** List of selected neuroimaging studies addressing cognitive dysfunctions in ALS.

	AUTHORS	NEUROIMAGING MODALITY	#ALS/CONTROL	METHOD/TASK	MAJOR FINDINGS IN NON-MOTOR REGIONS IN ALS
**REST**	Ma et al., 2015 [49]	fMRI	20/20	Functional network analysis	Within the hub regions, higher functional connectivity in the prefrontal cortex with the connectivity strength in the abnormal hub associated with clinical variables, was obtained
	Luo et al., 2012 [62]	fMRI	20/20	Functional connectivity	Alteration in low-frequency fluctuations in frontal areas was concluded
	Douaud et al., 2011 [63]	fMRI + DWI	25/15	Functional/structural connectivity	Increased functional connectivity spanning sensorimotor, premotor, prefrontal and thalamic regions associated with decreased structural connectivity, was obtained
	Agosta et al., 2013 [65]	fMRI	20/15	Functional network analysis	A divergent connectivity pattern in the default mode network (DMN), including decreased connectivity of the right orbitofrontal cortex and increased parietal connectivity, was associated with clinical and cognitive deficits
	Li et al., 2017 [67]	fMRI	21/21	Functional connectivity	Decreased functional connectivity in the left and right ventrolateral PFC, along with widespread and frequency-dependent FCS changes, provided evidence that ALS patients exhibit consistent impairments in extra-motor regions even at relatively early stages of the disease
	Fraschini et al., 2016 [69]	EEG	21/16	Functional network analysis	A significant group difference in MST dissimilarity and MST leaf fraction in the beta-band was found
	Kopitzki et al., 2016 [70]	fNIRS + DTI	31/30	Functional (hemodynamic) /structural connectivity	Anterior-temporal homotopic rs-FC was found to correlate with fractional anisotropy in the central corpus callosum (CC)
	Deligani et al., 2020 [59]	fNRIS + EEG	10/9	Functional connectivity	Increased fronto-parietal EEG connectivity in the alpha and beta bands, along with increased interhemispheric and right intra-hemispheric fNIRS connectivity in the frontal and prefrontal regions, was observed
	Nasseroleslam et al., 2019 [71]	EEG + MRI	100/34	Functional/structural connectivity	Increased EEG coherence between parietal–frontal scalp regions in the gamma band was observed. EEG signals associated with less extensively involved non-motor regions also showed enhanced structural connectivity on MRI
	Dukic et al., 2019 [72]	EEG+ MRI	74/47	Functional connectivity	Increased co-modulation in frontal regions (theta and gamma bands), along with decreased synchrony in the temporal and frontal regions (theta to beta bands), was observed
**TASK-BASED**	Riccio et al., 2013 [84]	EEG	8/NA	P300-BCI spelling	The temporal filtering capacity in the RSVP task was a predictor of both the P300-based BCI accuracy
	Pinkhardt et al., 2008 [85]	EEG	20/20	ERP/dichotic listening task	A distinct decrease in the fronto-precentral negative difference signal (Nd) was observed in ALS, along with increased processing of non-relevant stimuli, as reflected by the P3 component, suggesting a reduced focus of attention
	Shahriari et al., 2019 [86]	EEG	9/NA	P300-BCI spelling	BCI performance was positively correlated with a positive deflection in EEG amplitude around 220 ms at frontal mid-line locations
	Zisk et al., 2021 [90]	EEG	6/9	P300-BCI spelling	Latency jitter was significantly increased in participants with ALS
	Kasahara et al., 2012 [112]	EEG	8/11	ERD/motor imagery task	The ERD of ALS patients was significantly smaller
	Hosni et al., 2019 [113]	EEG	6/11	ERD/motor imagery task	Decreased ERD features were correlated with ALS clinical scores, specifically disease duration, bulbar, and cognitive functions
	Stanton et al., 2007 [115]	fMRI	16/17	Motor imagery task	Reduced activation during motor imagery was observed in ALS in the left inferior parietal region, as well as in the anterior cingulate gyrus and medial prefrontal cortex
	Borgheai et al., 2020 [127]	fNIRS	9/10	Functional network analysis /visuo-mental task	A more centralized frontal network organization was observed in the ALS group, with the most frequent network hubs showing an asymmetrical pattern predominantly localized in the right prefrontal cortex
	Shahriari et al., 2015 [136]	EEG	9/13	P300/BCI spelling	A significantly broader distribution of high-beta band connectivity, along with an absence of distinct connections in the directivity analysis, was observed in ALS patients
	Hammer et al., 2010 [138]	EEG	17/20	ERP/Go-NoGo task	An anteriorization of the NoGo-P3 was observed, a pattern that has been established as an index of impaired inhibitory function
	Proudfoot et al., 2017 [151]	MEG	11/10	Spectral power/motor task	A relative increase in beta-power was revealed in the frontal lobe and premotor regions

## 7. Correspondence Between Structural and Functional Neuroimaging and Other Cognitive Biomarkers of ALS

The relationship between structural and functional neuroimaging and other ALS disease biomarkers is an area that is still under investigation, but emerging studies have explored some associations. In a study of 64 ALS patients and 27 controls [156], analyses of structural imaging, diffusion-weighted imaging (DWI), and resting-state fMRI networks revealed a shared pathological pattern; the brain connections exhibiting the greatest structural damage corresponded directly to those showing the greatest functional impairment. Comparing 74 ALS patients with 47 age-matched healthy controls, Dukic et al. [72] reported cortico-cortical EEG connectivity (imaginary coherence [iCoh]) changes (beta-band in motor and delta-band in frontoparietal and frontotemporal channels) associated with structural degeneration as captured by MRI (within the grey matter volume of motor and frontal cortical channels), as well as the functional (ALSFRS-R) and cognitive scores. This study applied two-way ANOVA on all their EEG measures (spectral power and iCoh) that could categorize ALS patients versus healthy cohorts but did not result in a significant difference between patients’ subgroups based on genomic (C9ORF72) status. However, in their follow-up study [155], using clustering methods, the authors categorized four different ALS sub-phenotypes based on distinct EEG network features, where one of the clusters had the highest proportion of C9ORF72-positive patients. Similarly, in another ALS study [157], when graph theory analysis was applied to RSFC between MRI-based subcortical volumes, C9ORF72-positive patients with behavioral variant FTD (with and without comorbid ALS) showed reorganization of network hubs compared to those with ALS only (without behavioral variant FTD). Similarly, Geronimo et al. [158] reported that C9ORF72-positive ALS patients (though low in number: *n* = 4) exhibited greater suppression in visual evoked potential (VEP) and online P300 BCI performance compared to both control and ALS-only groups. A recent resting-state EEG study has also reported alpha and beta band power suppression in ALS patients associated with increased C9orf72 expression and reduced cortical thickness in parietal, occipital and temporal regions [159]. In another task-based EEG study [160] in 87 asymptomatic family members (AFMs) of patients with familial C9orf72 ALS, stimulus-locked evoked potentials were compared between 37 AFM individuals carrying the pathological repeat expansion (C9+) and 50 AFMs without expansion (C9−). Go and NoGo stimulus-locked N2 (180–350 ms) in C9+ AFMs showed an increased negative potential compared to the C9- cohort in the bilateral precuneus and superior parietal regions. Another recent resting-state MEG study [161] could classify symptomatic ALS, asymptomatic C9orf27 and SOD1 carriers compared to healthy controls. In their report, the symptomatic ALS group (*n* = 61) was characterized by widespread reduced beta power across temporal, frontal, and motor regions. This contrasted sharply with asymptomatic C9orf72 carriers (*n* = 16), who showed a mixed pattern of decreased beta power (occipital, temporal) and increased power in other bands (frontal theta, motor/occipital alpha), while asymptomatic SOD1 carriers (*n* = 12) exhibited no significant change in beta power. These findings indicate that functional neuroimaging biomarkers have the potential not only to distinguish patients from healthy controls but also to differentiate between genomic and non-genomic ALS subgroups.

It is noteworthy that there is potential to link neuroimaging biomarkers to cortical hyperexcitability in ALS [162,163,164]. This includes studies using transcranial magnetic stimulation (TMS) and neuroimaging to probe upper/lower motor neuron dysfunction or hyperexcitability. However, these studies, so far, have focused mainly on the primary motor cortex [162]. Thus, there is a critical need to broaden these neurophysiological investigations to capture the full spectrum of ALS as a multisystem disease. For instance, expanding TMS-EEG studies [163,164] to non-motor cortical regions, such as the prefrontal or parietal cortex, is crucial for identifying patterns of excitability dysfunction linked to common cognitive and behavioral deficits (e.g., FTD comorbidity). Furthermore, assessing excitability changes while subjects perform cognitive tasks (moving beyond the passive resting state) allows researchers to dynamically challenge the functional integrity of both motor and non-motor networks, providing more sensitive, active biomarkers for early cognitive decline and network breakdown that may be missed during simple rest.

## 8. Summary and Perspectives

ALS is diagnosed as a disease of progressive motor paralysis caused by degeneration of upper and lower motor neurons. However, from a genetic perspective, ALS is closely related to FTD, which has a hallmark of progressive degeneration of frontal cortical regions resulting in neurocognitive dysfunction and dementia. In reality, these related neurodegenerative conditions are heterogeneous in disease presentation, course and underlying neural system dysfunction. There is substantial evidence that patients with ALS also suffer neurocognitive dysfunction, as measured using both generalized clinical cognitive test batteries such as the MoCA and MMSE and batteries specifically designed for testing cognitive function in ALS patients such as the ECAS and ALS-CBS. Approximately one-third of people with ALS show these cognitive deficits, although it is as yet uncertain whether these deficits are progressive. This indeterminacy reflects, in part, the general limitations of such tests and, in particular, the difficulty of measuring cognitive function in the ALS population, i.e., against the backdrop of the progressive motor deterioration and the emotional and affective lability that accompanies this devastating fatal illness. Notwithstanding, the presence of cognitive dysfunction in ALS patients points to broader neural dysfunction beyond the motor system. The ability to sensitively measure this broader neural dysfunction would afford a more fundamental understanding of the pathophysiology underlying ALS while providing biomarkers to follow disease progression and use in testing potential therapeutic agents. EEG and fNIRS are proving valuable tools in investigating the broader neural dysfunction in ALS. These modalities reflect cortical synaptic activity at different and complementary spatial and temporal resolutions. Importantly, these neuroimaging modalities are ‘portable’, a significant consideration given the limited mobility of many with ALS. While there are currently no definitive neurophysiological signatures of ALS, emerging trends are summarized below.

Neural dysfunction measurable by EEG, fNIRS, and other imaging modalities is evident at the early stages of sensory processing, detected as abnormal ERP responses. This is particularly notable in the P300 and later ERP epochs, which reflect the initial integration of sensory signals into higher internal cognitive processing such as attentional systems. ERP responses in ALS patients are both diminished and temporally more variable, i.e., have higher jitter. Timing is critical to the integration of individual neuronal signals into functional networks. Thus, it is plausible that an increase in jitter, as directly measured in early sensory processing tasks, also contributes to the decay in broader network integrity, evident in analyses when subjects are performing more complex motor imaging tasks. Examples of the latter include the synchrony of neural activity across brain regions in the mu and beta frequency bands that occurs with motor imagery, ERD and ERS, which are consistently found to be abnormal in ALS patients. Deficits in neural synchrony across brain regions in ALS patients are beginning to be investigated using analytical techniques such as graph theory to both reflect and inform on higher-order functional neurocognitive deficits. These analyses point to the degradation of frontal cortical networks related to executive functioning, attention, and working memory manipulation. Frontal network dysfunction is a cardinal feature of FTD, and the finding of such dysfunction in ALS patients is consistent with putatively common underlying neurodegenerative processes, as suggested by the overlapping genetic underpinnings of these diseases. It is also noted that the neurophysiological studies indicating frontal network dysfunction in ALS suggest the neurocognitive domains instantiated by these networks bear further clinical investigation. Thus, while ALS is fundamentally a ‘motor’ disease, neurophysiology points to broader neural systems dysfunctions affecting multiple cognitive domains. This understanding has consequences in caring for the needs of ALS patients as well as developing therapeutics to treat their underlying neural dysfunctions.

A corollary to an understanding of the neurophysiological dysfunctions underlying ALS is the development of potential biomarkers to follow disease progression and use in clinical trials of ALS therapeutics. In this regard, the hierarchical nature of the neurophysiological abnormalities in people with ALS may be used to clinical advantage. Combined measures may be particularly useful in quantifying potential efficacy in clinical trials of therapeutic interventions. To the extent to which distortions in electrophysiological and/or hemodynamic features, in temporal, spectral, connectivity, or network domains, reflect the underlying neurodegenerative process, therapies that can be shown to impact these features in early clinical testing may be promoted for clinical evaluation of efficacy against the broader array of ALS symptoms. Finding therapeutics that impact the cognitive and non-motor functional decay in ALS may be a sensitive and accessible avenue towards comprehensive therapies that also address the more aggressive motor degeneration in the disease.

## 9. Limitations, Gaps, and Future Directions

Neuroimaging studies dissecting cognitive dysfunctions in ALS are characterized by considerable methodological and analytical variability, limiting generalizability, comparability, interpretability, and the development of reliable biomarkers that can be used in practical clinical settings. Key challenges include small sample sizes, limited use of cognitive task–based paradigms, insufficient integration of multimodal neuroimaging data (e.g., combining EEG, fNIRS, MRI, or DTI to capture complementary temporal–spatial signatures), variability in preprocessing pipelines, lack of standardized processing metrics, and inadequate age- and disease-severity matching across cohorts. Furthermore, biological and genetic markers are often under-incorporated in study designs, resulting in reduced ability to account for their influence. Compounding this, many studies insufficiently control for additional confounding factors, such as medications and comorbidities including depression, anxiety, sleep disturbances, and chronic pain, all of which can independently affect cortical oscillatory activity. Moreover, measures of functional connectivity vary widely across studies due to substantial differences in the choice of connectivity metrics and frequency-specific analyses. The aforementioned issues are often compounded by the limited spatial and temporal validity of neuroimaging modalities. For example, EEG source localization for subcortical structures can render inferences about cortico-subcortical connectivity particularly uncertain. Consequently, EEG-derived connectivity findings, especially those involving deep brain networks, are often inconsistent, reflecting methodological constraints rather than true pathophysiological differences. The inherently low spatial resolution of EEG and contamination of frontal channels by facial muscle artifacts (e.g., eye movements, blinks, or jaw tension) further complicate analysis and data interpretation. Therefore, adopting integrative approaches, such as utilizing network analysis based on multimodal neuroimaging (e.g., combining EEG with fNIRS, which offers high frontal capturing potential), especially while patients are performing cognitive tasks, is increasingly essential for overcoming methodological limitations in neuroimaging studies of ALS. The concurrent recording of electrophysiological (EEG) and hemodynamic (e.g., fNIRS) signals allows capture of both temporal dynamics and spatially targeted cortical activation, increasing the likelihood of detecting disease-relevant network dysfunction while reducing reliance on any single modality’s limitations. Indeed, methodological reviews highlight the growing use of such dual-modal systems, classify common integration strategies (e.g., EEG-informed fNIRS, fNIRS-informed EEG, parallel analyses), and outline their advantages and challenges [165,166,167,168]. Recent studies show that multimodal, graph-based frameworks are increasingly being applied in neurorehabilitation and BCI studies, demonstrating both their feasibility and potential to uncover clinically meaningful patterns of network reorganization [60,169,170,171]. Adopting graph-theoretical network analysis on multimodal data may thus offer more robust biomarkers. By integrating spectral, hemodynamic, and connectivity features into indices of global and local network organization (e.g., efficiency, modularity, hub disruption), such approaches can capture disease-related alterations in large-scale brain networks that are less sensitive to modality-specific noise or methodological variability than pairwise connectivity metrics.

The aforementioned methodological limitations, together with intrinsic heterogeneity of ALS, further contribute to divergent findings reported across studies. ALS exhibits pronounced variability in site of onset, balance of upper- and lower-motor-neuron involvement, rate of progression, and the presence of cognitive or behavioral impairment, each of which is likely associated with distinct patterns of cortical and network-level dysfunction. Small, cross-sectional cohorts often fail to capture this phenotypic diversity, leading to overrepresentation of specific disease subtypes and increased inter-study variability. In this context, analyzing patients based on distinct sub-categories can offer a promising and necessary framework to effectively cluster the heterogeneous clinical symptom domains and dissect the unique underlying neural mechanisms that drive disease variability. By combining detailed clinical phenotyping, longitudinal designs, multimodal imaging, and network-level analytical approaches, future studies can better disentangle true ALS-related electrophysiological signatures from variability driven by analytic choices, source-localization limitations, and the underlying biological heterogeneity of the disease.

## Figures and Tables

**Table 1 cells-15-00037-t001:** Gene-specific ALS phenotypes.

Gene	Frequency	Typical Age of Onset	Phenotypic Tendencies	Comments
SOD1	~20% of fALS	40–60	Limb-onset; variable progression depending on mutation	Some variants are aggressive, others slow
C9ORF72	~34% of fALS	40–70	Bulbar-onset more common; psychiatric symptoms; ALS-FTD overlap	Immune and autophagy dysfunction
TARDBP (TDP-43)	3–5% of fALS	50 s	Slow progression except certain variants (e.g., G376D)	RNA metabolism defects
FUS	4–5% of fALS	Juvenile–adult	Aggressive, early onset; often cognitive symptoms	Nuclear transport defects

## Data Availability

No new data were created or analyzed in this study.

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
