# Peer review of "A Comprehensive Overview of Neurophysiological Correlates of Cognitive Impairment in Amyotrophic Lateral Sclerosis"

_cells, 2025, doi:10.3390/cells15010037_

Round 1
Reviewer 1 Report
Comments and Suggestions for Authors
Achorn et al. comprehensively review the current tools and methodologies to evaluate cognitive impairment in people suffering from amyotrophic lateral sclerosis (ALS), focusing on neuroimaging techniques. ALS is commonly associated with motor dysfunction resulted from the preferential loss of upper and lower motor neurons. However, around 30% of patients present with cognitive impairment at the time of diagnosis, which demonstrates the significance and impact of this problem for the quality of life of the patients and their caregivers.
This reviewer has no major issues with the manuscript, which is superbly written and organized. However, it is curious that the work of Ringholz et al. (2005), which is foundational as to changing perception of ALS as a purely motor disease, is not among the References.
Although this may be a bit outside the scope of this work, it would be informative if the authors could document efforts in the field to correlate neuroimaging biomarkers of cognitive impairment with commonly used biomarkers of neuronal/axonal degeneration, muscle electrophysiological changes, or neuroinflammation.
The following minor issues need to be addressed:
Line 44: “It is the most common neurodegenerative disease to affect middle aged individuals” Please double check the accuracy of this statement. PD and AD may be more prevalent than ALS at this age.
Line 98: Human gene abbreviations should be capitalized and italicized throughout the manuscript.
Author Response
See attached response to reviewers

Reviewer 2 Report
Comments and Suggestions for Authors
This manuscript presents a comprehensive and up-to-date review of cognitive impairment in ALS, focusing on neurophysiological correlates derived from multimodal imaging (EEG, fNIRS, fMRI, PET) and electrophysiological analyses. The review demonstrates impressive breadth, covering clinical, genetic, structural, and functional dimensions of ALS and its overlap with FTD. The writing is clear and technically accurate, and the topic is highly relevant given the increasing recognition of ALS as a multisystem disorder. However, the paper would benefit from stronger critical synthesis, conceptual integration, and clearer methodological framing. The current version reads largely as a descriptive compilation rather than a mechanistic or hypothesis-driven synthesis.
Major concerns as following:
1, The paper does not state how the literature was collected, screened, or prioritized. No description of inclusion criteria, search databases, or timeframe. The authors should add a brief Methods subsection explaining the search strategy or state explicitly that this is a narrative review rather than systematic.
2, Many sections (e.g., “Clinical Features” and “Genetics”) repeat textbook material and consume too much space relative to the article’s main goal (neurophysiological correlates). The neurophysiology sections, while informative, mostly summarize previous findings without identifying contradictions, gaps, or emerging trends. The authors should condense basic background and devote more text to critical comparisons among studies (e.g., why EEG and fNIRS findings diverge; how cognitive decline correlates inconsistently across modalities).
3, The review lists numerous imaging results (EEG power changes, network connectivity, PET hypometabolism, etc.) but rarely integrates them into a unifying pathophysiological model. There is potential to link these findings to known molecular mechanisms (e.g., TDP-43 pathology, cortical hyperexcitability, neuroinflammation, network disconnection). The authors should include a conceptual figure or paragraph synthesizing how neurophysiological changes map onto ALS–FTD spectrum pathology.
4, Several statements imply consensus (“EEG and fNIRS are proving valuable tools”) even when cited studies show inconsistent or opposite results. The authors need to explicitly acknowledge discrepancies (e.g., hyperconnectivity vs. hypoconnectivity reports; progressive vs. stable cognitive trajectories) and discuss possible technical or cohort-related causes.
5, The article ends by suggesting EEG/fNIRS as biomarkers but does not evaluate their specificity, reproducibility, or clinical applicability (e.g., signal variability, standardization issues, portability vs. sensitivity trade-offs). The authors should expand the “Summary and Perspectives” with a subsection on clinical and translational potential, including standardization barriers, integration with genetic or fluid biomarkers and utility in clinical trials (endpoint feasibility, longitudinal reproducibility).
6, The introduction and clinical/genetic overview (Sections 1–2) occupy a large portion (>40%) of the manuscript, while the main focus, neurophysiological correlates, appears later. The authors are suggested to shorten the background to two concise pages and move detailed ALS phenotypes to a table or appendix.
7, The review rarely comments on limitations of the cited studies (small sample sizes, heterogeneous ALS subtypes, varying cognitive scales, lack of longitudinal control). The authors should add a “Limitations of current evidence” section summarizing methodological weaknesses across studies.
Minor defects:
1, The authors should ensure consistent use of abbreviations (ECAS, fNIRS, ERD/ERS, RSFC).
2, The authors can include a schematic of ALS cognitive networks and a table summarizing EEG/fNIRS markers correlated with cognition.
3, Some citations are outdated or redundant (e.g., 2010–2015 imaging studies could be replaced by 2022–2024 meta-analyses). The authors should verify numerical consistency between in-text citations and reference list.
4, Several sentences are overly long and complex; The authors can consider splitting for clarity and flow.
Author Response
See attached responses to reviewers

Round 2
Reviewer 2 Report
Comments and Suggestions for Authors
All concerns were addressed properly. No further concerns. Suggest accepting it